# Tellurium Doping Inducing Defect Passivation for Highly Effective Antimony Selenide Thin Film Solar Cell

**DOI:** 10.3390/nano13071240

**Published:** 2023-03-31

**Authors:** Guojie Chen, Xiangye Li, Muhammad Abbas, Chen Fu, Zhenghua Su, Rong Tang, Shuo Chen, Ping Fan, Guangxing Liang

**Affiliations:** 1Shenzhen Key Laboratory of Advanced Thin Films and Applications, Key Laboratory of Optoelectronic Devices and Systems of Ministry of Education and Guangdong Province, College of Physics and Optoelectronic Engineering, Shenzhen University, Shenzhen 518060, China; 2060451014@email.szu.edu.cn (G.C.); williamlixiangye@gmail.com (X.L.); abbas.cssp@hotmail.com (M.A.); zhsu@szu.edu.cn (Z.S.); fanping@szu.edu.cn (P.F.); lgx@szu.edu.cn (G.L.); 2School of New Energy and Energy Conservation and Environmental Protection Engineering, Foshan Polytechnic, Foshan 528137, China; rongtang@fspt.edu.cn

**Keywords:** Sb_2_Se_3_, Te doping, defects passivation, solar cell, efficiency

## Abstract

Antimony selenide (Sb_2_Se_3_) is emerging as a promising photovoltaic material owing to its excellent photoelectric property. However, the low carrier transport efficiency, and detrimental surface oxidation of the Sb_2_Se_3_ thin film greatly influenced the further improvement of the device efficiency. In this study, the introduction of tellurium (Te) can induce the benign growth orientation and the desirable Sb/Se atomic ratio in the Te-Sb_2_Se_3_ thin film. Under various characterizations, it found that the Te-doping tended to form Sb_2_Te_3_-doped Sb_2_Se_3_, instead of alloy-type Sb_2_(Se,Te)_3_. After Te doping, the mitigation of surface oxidation has been confirmed by the Raman spectra. High-quality Te-Sb_2_Se_3_ thin films with preferred [hk1] orientation, large grain size, and low defect density can be successfully prepared. Consequently, a 7.61% efficiency Sb_2_Se_3_ solar cell has been achieved with a *V*_OC_ of 474 mV, a *J*_SC_ of 25.88 mA/cm^2^, and an FF of 64.09%. This work can provide an effective strategy for optimizing the physical properties of the Sb_2_Se_3_ absorber, and therefore the further efficiency improvement of the Sb_2_Se_3_ solar cells.

## 1. Introduction

Antimony selenide (Sb_2_Se_3_) has become a promising light-harvesting material, thanks to the advantages of high earth abundance, low toxicity, superior photoelectric properties, and a high light absorption coefficient [1,2,3,4,5]. Sb_2_Se_3_ as a binary compound has high stability and fixed stoichiometry. Moreover, Sb_2_Se_3_ has a band gap of ~1.1 eV, suitable for single-junction solar cells. According to the Shockley-Queisser limit, this small band gap determined a theory power conversion efficiency (PCE) exceeding 32% for a single-junction solar cell [6]. In addition, Sb_2_Se_3_ owns a 1D crystal structure composed of [Sb_4_Se_6_]_n_ nanoribbons with strong covalent connections along the (001) axis and van der Waals (vdW) forces along the (100) and (010) axes [7,8,9]. Hence, it is required to induce the favorable [hk1] orientation in the Sb_2_Se_3_ thin film, which is advantageous for the photogenerated carrier to transport and results in an improvement of PCE. 

It is widely known that doping is an effective strategy to optimize the property of semiconductors [10]. In the family of V_2_VI_3_ (V = Sb, Bi; VI = Se, Te), Sb_2_Te_3_ illustrated a rhombohedral crystal structure while Sb_2_Se_3_ exhibited an orthorhombic crystal structure. The introduction of tellurium (Te) in the Sb_2_Se_3_ has been found to effectively adjust the electrical and optical properties [11,12]. Ma et al. [13] reported Te-doped Sb_2_Se_3_ thin film synthesized by spin-coating, which indicated that Te-doping can successfully minimize the deep-level defects in the Sb_2_Se_3_ thin film by optimizing the change in Sb/Se atomic ratios. Rahman et al. [14] studied the Te-doped Sb_2_Se_3_ thin film obtained by thermal evaporation from the bulk powder prepared using the melt quenching technique. A reduction in surface oxidation may be detected in Sb_2_Se_3_ thin films, and the DC conductivity of the Te-doped Sb_2_Se_3_ thin film is in the region of 10^−7^ Ω^−1^ cm^−1^ at 308.5 K, which is ten times more than the conductivity of pristine Sb_2_Se_3_ films that had previously been reported. Ren et al. [15] found that Te atoms were accommodated into the gap of [Sb_4_Se_6_]_n_ ribbons, and the Te-doped Sb_2_Se_3_ films oriented along the plane (221), which is beneficial for carrier transport. Hence, it is necessary to solve the issues that limit the further improvement of device performance, such as surface oxidation, deep-level defect, and the preferred [hk1] orientation.

In this work, we have successfully fabricated the Te-doped Sb_2_Se_3_ thin film by a co-post-selenization treatment. By carefully modifying the temperature and the weights of the Te source, the Se/Sb atomic ratios and the bulk defect of Sb_2_Se_3_ can be effectively optimized. According to the results, we found that the Te-doping tended to form Sb_2_Te_3_-doped Sb_2_Se_3_, instead of the alloy type Sb_2_(Se,Te)_3_. Besides, proper Te doping can effectively alleviate the surface oxidation phenomenon and passivate the defects of the absorber leading to further enhancement of the absorber quality. Consequently, the solar cell device with the architecture of Mo/Te-Sb_2_Se_3_/CdS/ITO/Ag exhibits the best performance of 7.61% with an open-circuit voltage (*V*_OC_) of 474 mV, short-circuit current density (*J*_SC_) of 25.88 mA cm^−2^ and fill factor (FF) of 64.09%.

## 2. Materials and Methods

### 2.1. Te-Sb_2_Se_3_ Thin Film Deposition

Firstly, a radiofrequency (RF) magnetron sputtering deposition method was used to obtain the Sb metallic precursors thin films. Before the depositions, Mo-coated glass substrates were ultrasonically cleaned with detergent solution, deionized water, and ethanol for 20 min, respectively. Before beginning to deposit the Sb thin film, the background pressure of the sputtering system was reduced to 8.0 × 10^−4^ Pa. At a flow rate of 35 sccm high-purity (>99.999%) Ar gas was fed into the sputtering chamber. (Figure 1a) During the sputtering process, the power and duration were fixed at 30 W and 45 min, respectively. The working pressure was kept at 1.8 Pa. To obtain the Te-Sb_2_Se_3_ thin film, The precursor thin films were subsequently put into a vacuum tubular furnace for post-selenization treatment (Figure 1b). The selenization temperature and the quantity of Te powder were adjusted from 440 °C to 480 °C and from 0 g to 0.2 g, respectively, to study the influence of selenization parameters on device behavior. A total of 0.4 g of selenium powder was applied to each sample for the selenizaiton process. The temperature of the selenium source and tellurium source is controlled by two temperature zones. Before the selenization the tubular furnace was evacuated to a low background pressure. After that, the furnace was filled with high purity Ar (>99.999%), and the working pressure was maintained at 7 × 10^4^ Pa throughout the annealing process. Thereafter, at a rate of 20 °C/min, the furnace temperature was increased to 440 °C for the Te source and 420 °C for the Se source, respectively. For each sample, the selenizing time was set at 15 min. After the heating cycle was over, the furnace automatically cooled to room temperature.

### 2.2. Te-Sb_2_Se_3_ Thin-Film Solar Cell Preparation

Chemical bath deposition (CBD) was used to obtain the cadmium sulfide (CdS) buffer layer following the post-selenization procedure. (Figure 1c). The CdS layer was then coated with an indium tin oxide (ITO) layer using magnetron sputtering for 25 min at a DC voltage of 120 W (Figure 1d). The active device area was 0.12 cm^2^ after Ag conductors were thermally evaporated onto the ITO surface to create metallic contact. Figure 1f shows an overall schematic layout of the device preparation procedure and the device configuration.

### 2.3. Device Characterization

The scanning electron microscope (SEM, SUPRA 55, Zeiss, Jena, Germany) was introduced to characterize the Te-Sb_2_Se_3_ thin film morphology. The crystal orientation of the Te-Sb_2_Se_3_ films was examined using X-ray diffraction (XRD, Ultima-iv, Rigaku, Tokyo, Japan, CuK radiation under operation circumstances of 40 kV and 40 mA from 10° to 60°). The excitation frequency used to produce the Raman spectra (Renishaw, Wotton-under-Edge, UK) was 532. A class AAA solar simulator (SCS1011, Zolix, Beijing, China) was used to measure the current density-voltage (*J*-V) profiles of the devices under illumination of 100 mW cm^2^ AM 1.5 G light at room temperature. A Zolix SCS101 system and a Keithley 2400 source meter were used to quantify the external quantum efficiency (EQE) spectra. A parameter analyzer was used to conduct capacitance-voltage (C-V) and drive-level capacitance profiles (DLCP) tests (Keithley, 4200A-SCS, Cleveland, OH, USA). The admittance spectroscopy (AS) measurement was performed using a Victor digital LCR meter in the frequency band of 100–1 MHz and 180–330 K (Lake Shore, 325 cryogenic temperature controller, Carson, CA, USA). A CHI600E Electrochemical Workstation (Shanghai, China) (was used to perform the electrochemical impedance spectroscopy (EIS). 

## 3. Results and Discussion

The Te-Sb_2_Se_3_ thin films selenized with the different weights of Te powder were denoted as T0, T1, T2, and T3, respectively. T0 corresponds to the weight of 0 g Te powder, T1 corresponds to the weight of 0.1 g Te powder, T2 corresponds to the weight of 0.15 g Te powder and T3 corresponds to the weight of 0.2 g Te powder. Besides, the Te-Sb_2_Se_3_ thin films selenized under different temperatures were denoted as T2-440 °C, T2-460 °C, and T2-480 °C, respectively, and the pristine Sb_2_Se_3_ thin film was denoted as T0-420 °C. Several carefully planned experiments were performed to gain insight into the connection between the Te source temperature and the thin film quality. Figure 2a–c depicted the SEM morphology of the pristine sample and the Te-doped sample under the same Se source temperature but with the various Te source temperature. As can be seen, the T0-420 °C, T2-440 °C, and T2-460 °C thin films showed a similar morphology. With the increased temperature of the Te source, the crystal grain size also presented an increasing trend. However, pinholes appeared on the surface of the T2-480 °C sample (Figure 2d). The film significantly deteriorated probably because the nucleation and crystal growth processes had changed, leading to the formation of apparent holes. The particle size distribution versus the fine frequency histograms were depicted in Figure 3. After Te doping, the T2-460 °C thin films revealed a steady improvement in compactness and uniformity as the temperature was raised from 440 to 460 °C, which was accompanied by a corresponding rise in average grain size from 1.02 m to 1.28 m. Nevertheless, when the temperature increased to 480 °C, some pinholes appeared. Under a high temperature, Se is simpler to evaporate out during the selenization process compared with Te, leading to the generation more pinholes. Subsequently, the Te and Se source temperatures were fixed at 460 °C and 420 °C, respectively. The surface SEM pictures of the Sb_2_Se_3_ thin films with various weights of Te powder during the selenization treatment are illustrated in Figure 2e–h. All the as-obtained films displayed a relatively dense surface morphology. As depicted in Figure 3e–f, the average grain size of the T0 sample is 1.3 μm. Once the Te with weight of 0.1 and 0.15 g were added during the selenization treatment, the average grain size increased to 1.31 and 1.36 μm, respectively. However, when the weight of Te increased to 0.2 g, the average grain size decreased to 1.20 μm. The grain growth in this instance was presumably prevented by more Te being concentrated at the grain boundaries.

The introduction of the X-ray diffraction (XRD) characterization was conducted to confirm the crystal structure of the as-obtained thin films. Figure 4 displayed the XRD patterns of the pure sample and the Te-doped samples at various weights and temperatures. It is clear that all of the sample diffraction peaks matched the typical ortho-rhombic Sb_2_Se_3_ diffraction patterns well (JCPDS 15-0861). Figure 4b showed the detailed analysis of enlarged XRD patterns. There are additional diffraction peaks indexed in the Te-doped Sb_2_Se_3_ thin films, which was in good agreement with the Sb_2_Te_3_ patterns (JCPDS 45-1229). Once we introduced the Te source during the selenization process, an additional Sb_2_Te_3_ phase appeared, rather than the peak of Sb_2_Se_3_ shifting. Besides, when increasing the weight of Te, the XRD patterns exhibited the same peak position (Figure 4c). This observation proved that increasing the reaction temperature and weight does not lead to the Te atoms substituting for the Se atoms in the lattice. It means that under the competitive reaction between the Te and Se during the selenization process it tended to form Sb_2_Te_3_-doped Sb_2_Se_3_, instead of the alloy-type Sb_2_(Se,Te)_3_. Moreover, it is notable that the [221] and [211] diffraction peaks intensity increased before decreasing, no matter whether increasing the reaction temperature or increasing the weight of Te. However, if the temperature and the weight of Te were too high and/or too much, e.g., 480 °C and 0.2 g, there was a slight decrease in the intensity of the diffraction peak, which could be explained by the formation of the Sb_2_Te_3_ phase that prevented the grain growth. Figure 4e,f demonstrated the Raman images of the thin films under different temperatures and weights of Te. As depicted in the Raman spectra, There were two obvious peaks centered around 189 and 210 cm^−1^, which are frequently ascribed to the Sb-Se-Sb bending vibration of Sb_2_Se_3_ [16,17]. The stabilization of the orthorhombic phase has been further confirmed, which is in accordance with the XRD results. Except that, an additional peak that appeared at 253 cm^−1^ was more closely related to α-Sb_2_O_3_ [18]. After introducing Te during the selenization, the peak around 253 cm^−1^ disappeared, and the peak intensity for Te-doped Sb_2_Se_3_ thin film decreased with higher insertion. Moreover, the energy-dispersive X-ray spectroscopy (EDS) was introduced to measure the element composition of the absorber layers. Such a situation may be due to the mitigation of surface oxidation or the tunning of the (Se/Sb) ratio (Table 1 and Table 2). Interestingly, it can be shown that the Sb_2_O_3_ peak got reduced at 460 °C and completely disappeared at 480 °C. A similar trend can be noticed in Figure 4e with increasing the weight of the Te source, which indicated that a lower temperature and weight for the Te source are insufficient to occupy the Sb sites during the selenization process.

Overall, a suitable selenization temperature and doping weight were crucial for obtaining high-quality Sb_2_Se_3_ thin film via carefully optimized selenization parameters. On the one hand, the introduction of Te can effectively induce the [hk1] orientation of the Sb_2_Se_3_ thin films, leading to higher carrier transport and extraction. On the other hand, harmful surface oxidation can be effectively mitigated after Te inclusion. Ultimately, the temperature and weight of the Te source were set at 460 °C and 0.15 g, respectively, while the temperature of the Se source was set at 420 °C. These values were determined to be the optimal selenization parameters.

To investigate the photovoltaic properties of the Te-doped Sb_2_Se_3_ thin films, Mo/Te-Sb_2_Se_3_/CdS/ITO/Ag was used in the fabrication of the devices. The optimized films and pristine films were used for comparison. Moreover, for clarity of description, the two samples were denoted as T0 and T2, respectively. The current density-voltage (*J*-V) curves of the devices were characterized under simulated AM 1.5 sunlight, and Table 3 shows the detailed photovoltaic properties of the two devices. The pristine Sb_2_Se_3_ solar cell exhibited a PCE of 6.42% with a *V*_OC_ of 460 mV, a *J*_SC_ of 23.15 mA/cm^2^, and an FF of 60.86%. While after Te-doping, the T2 device exhibited a much superior performance with a PCE of 7.61%, along with a *V*_OC_ of 474 mV, a *J*_SC_ of 25.88 mA/cm^2^, and an FF of 64.09% (Figure 5a). Notably, the enhancement of the performance after Te-doping could be mainly due to the synergetic development of the photovoltaic parameters, confirming that Te-doping may enhance the efficiency of the carrier transport, leading to better thin films. Besides, Figure 5c–f depicted the statistics box diagrams of performance parameters from these two devices, demonstrating the excellent repeatability of device fabrication.

To further analyze the photo responses of the devices, the external quantum efficiency (EQE) spectra of both devices was conducted, as displayed in Figure 5b. Compared with the virgin device, the T2 device exhibited a much higher response in the wavelength region from 500 nm to 900 nm, which is not only corresponding to better carrier transport and extraction leading to a higher *J*_SC_ but also suggests fewer recombination losses within both the bulk and the interface. The integrated *J*_SC_ values of the devices were 20.17 and 23.13 mA/cm^2^, respectively, which are consistent with the values obtained from the *J*-V curves. 

Capacitance-voltage (C-V) and deep-level capacitance profiling (DLCP) tests were performed to assess the defect density of the Sb_2_Se_3_ absorber layers more clearly and the combination properties of the solar cells. Figure 5g presented the pattern produced by C-*V* and DLCP. Information on interface defects could be directly obtained from the differences in doping density between the CV and DLCP data. The plots of *N_CV_* and *N_DLCP_* against the depth x can be calculated based on the following equation [19]:
(1)
$$N_{CV}=\frac{-2{ɛ}_{r,n}N_{D}}{\left(\frac{d\left(\frac{1}{C^{2}}\right)}{dV}\right)qA^{2}{ɛ}_{0}{ɛ}_{r,n}{ɛ}_{r,p}N_{D}+2{ɛ}_{r,p}}$$


(2)
$$N_{DLCP}=-\frac{C_{0}^{3}}{2q{ɛ}_{0}{ɛ}_{r,p}A^{2}C_{1}}\;\;\;\;$$


(3)
$$x=\frac{{ɛ}_{0}{ɛ}_{r,p}A}{C_{0}}\;\;$$

where *C*_0_ and *C*_1_ are two quadratic fitting parameters generated from the DLCP data; *ε_r,n_* and ε_r,p_ are the relative permittivity of CdS and Sb_2_Se_3_, respectively; *A* is the device area and; *N_D_* is the doping density of CdS. The interface defect density of the T0 device was 4.05 × 10^16^ cm^−3^, while after Te-doping, the interface defect density decreased to 8.45 × 10^15^ cm^−3^, indicating that there were fewer surface defects in Te-doped Sb_2_Se_3_ absorber. It is noteworthy that the carriers could be difficult to capture as a recombination center, but more easy to transport at the interface with the decreasing interface defect density, leading to much superior device performance [20]. In addition, the depletion width of the T2 device (325 nm) is obviously larger than that of the T0 device (278 nm). The collection and extraction of the charge carrier benefit from the wider depletion width [21]. As revealed in Figure 5h, the built-in voltage (*V*_bi_) of the T0 device is 0.42 V, and it increased to 0.69 V for the T2 device. The introduction of the Te enlarged the *V*_bi_ of the device forming an enhanced built-in voltages, supporting the improvement of the *V*_OC_ [22]. We used electrochemical impedance spectroscopy (EIS) measurements at frequencies ranging from 1 Hz to 1 MHz to shed light on the carrier recombination mechanism. The Nyquist curves were depicted in Figure 5i, and the inset graph demonstrates the corresponding fitted curves using an equivalent electrical circuit. Where *R_s_* represents internal resistance, and *R_rec_* represents recombination resistance [23,24]. Strikingly, the *R_rec_* increases from 10.9 kΩ to 37.3 kΩ after the introduction of Te-doping, implying that the severe carrier accumulation and recombination at the interface have been effectively passivated, thus providing a higher photovoltaic performance [25].

The dark *J*-V characterization was conducted to scrutinize the intrinsic electrical behaviors of the devices, as illustrated in Figure 6. A single exponential diode equation can be utilized to fit the shunt conductance (G), series resistance (R_S_), diode ideality factor (A), and reverse saturation current density (*J*_0_) [26,27].

(4)
$$J=J_{0}exp\left[\frac{q}{AkT}\;\left(V-JR\right)\right]+GV-J_{L}\;\;$$


Compared with the two devices, a lower reverse current density is found in the T2 device, suggesting that fewer carriers were trapped by defect recombination centers and more carriers successfully swept out of the device. As demonstrated in Figure 6b, the R-value is determined to be 6.08 and 4.58 Ω cm^2^ for T0 and T2 devices, respectively, and the obtained values of A were 1.56 and 1.32, respectively. The smaller value of A implies that the space-charge region (SCR) recombination has been suppressed effectively. Additionally, it is discovered that the *J*_0_ value of the T2 device is 5.0 × 10^−5^ mA/cm^2^, which is lower than that of the T0 device (1.6 × 10^−5^ mA/cm^2^). After Te-doping, the optimized absorber possesses a better quality, as well as a better PN junction quality, leading to a higher PCE. Finally, to better understand the thin films defect states, we introduced the standard space charge limited current (SCLC) model. Figure 6e,f illustrated the logarithmic *J*-V curves, which can be sectioned into three regimes: the ohmic region at low voltages (exponent n = 1), the trap-filled limit (TFL) region at intermediate voltages (*n* > 3), and the trap free child region at high voltages (*n* > 2). When the bias voltage exceeds the kink point, which corresponds to the trap-filled limit voltage, the injected carrier fills the trap states in the TFL region (*V_TFL_*). [28] Then the defect density of the Sb_2_Se_3_ thin film can be determined using the *V_TFL_* value according to the following equation [5]:
(5)
$$N_{trap}=\frac{2\varepsilon_{0}\varepsilon_{r}V_{TFL}}{qL^{2}}$$

where *L* represents the thickness of the Sb_2_Se_3_ thin film, q represents the elementary charge, *ε_r_* represents the permittivity of the Sb_2_Se_3_ and *ε*_0_ represents the vacuum permittivity. The defect densities were calculated to 2.06 × 10^14^ cm^−3^ and 1.86 × 10^14^ cm^−3^, respectively. The SCLC characterization confirmed that the Sb_2_Se_3_ films with larger grain sizes and benign orientation are capable of reducing the inter-defect density, echoing the previous analysis.

To further determine the defect characters of the Sb_2_Se_3_ films, the admittance spectroscopy (AS) measurements were applied. Figure 7a,b demonstrated the capacitance–frequency (C–f) curves for the temperature range of 180–340 K at 10 K intervals. The capacitance at low frequency indicates the response of the sum of the free carrier and deep defect, while the capacitance at high frequency represents the response of the free carrier density, according to the Kimerling model [29,30]. Compared to the T0 device, the capacitance of the T2 device showed less change as a function of frequency. Te-doping may be able to successfully passivate defects to improve the quality of the absorber layer because the more significant the frequency dependency of the capacitance, the greater the defect densities in the absorber. The angular frequency point at the maximum of the *ωdC*/*dω* plot was used to calculate the Arrhenius plots for the inflection points in the two AS results, which are exhibited in Figure 7c,d. Then, using the following equation, we determined the defect activation energies (*E_a_*) based on the slopes of the Arrhenius plots [31,32].

(6)
$$\omega_{0}=2\pi \nu_{0}T^{2}\exp(\frac{-E_{a}}{kT})$$

where *E_a_* is the defect electronic state level and the *ω*_0_ is the step frequency, *υ*_0_ is the pre-exponential factor. The *E_a_* values of 473 and 430 meV are determined for T0 and T2 devices, respectively, indicating that they are of the same type of defect. Based on a previous report, the Sb_2_Se_3_ thin films deposited via a post-selenization are slightly Se-rich. Under Se-rich conditions, the dominant defects of the Sb_2_Se_3_ thin films are antimony vacancy (V_Sb_) and selenium antisite (Se_Sb_) defects [33]. As a function of frequency, only one capacitance step can be recognized. It is difficult to differentiate them from each other. Thus, we tentatively considered both these defects (V_Sb_ and Se_Sb_) to be the dominant defects in our devices. It is well known that defects with higher *E_a_* values are more likely to become recombination centers, which lowers device performance. The defect densities of the two devices were fitted with a Gaussian curve employing following equation to provide relative quantitative data about the defect [34,35].

(7)
$$E\left(\omega \right)=\mathrm{kT}\ln\left(\frac{2\pi \nu_{0}T^{2}}{\omega }\right)\;\;\;\;\;\;$$


(8)
$$\;N_{t}\left(E\left(\omega \right)\right)=-\frac{V_{d}}{q\omega }\frac{dC}{d\omega }\frac{\omega }{\mathrm{kT}}\;\;\;\;\;$$

where *V_d_* represents the built-in voltage of the P-N junction, *E* represents the energetic gap between the defect energy level and the CBM or VBM, and *N_t_* represents the defect density. Figure 7e,f depicted the defect density of the T2 device (2.70 × 10^16^ cm^−3^), which was considerably less than that of the T0 device (1.47 × 10^17^ cm^−3^). In this regard, Te doping was essential in passivating the Sb_2_Se_3_ bulk defects.

## 4. Conclusions

In conclusion, our work has investigated the impact of Te doping on the functionality of the device. A two-step deposition process (sputtering and post-selenization) was introduced to prepare high-quality Te-Sb_2_Se_3_. The introduction of Te in the selenization process tended to form Sb_2_Te_3_-doped Sb_2_Se_3_, instead of the alloy-type Sb_2_(Se,Te)_3_, due to the competitive reaction between Te and Se. By a fine-tuning of the selenization parameters, high-quality, large grain size, and benign orientation Te-Sb_2_Se_3_ thin films can be successfully fulfilled. The characterizations and analyses have demonstrated that Te doping in Sb_2_Se_3_ not only enhances the quality of the Sb_2_Se_3_ absorber, leading to mitigation of surface oxidation, but also induces the thin film to present a favorable [hk1] orientation, leading to a better carrier transport. Finally, the champion Te-doped Sb_2_Se_3_ device exhibited a PCE of 7.61% with a *V*_OC_ of 474 mV, a *J*_SC_ of 25.88 mA/cm^2^, and an FF of 64.09%. This study demonstrated an effective doping strategy, which can significantly enhance the Sb_2_Se_3_ physical properties, providing a helpful direction for the creation of Sb_2_Se_3_ solar cells.

## Figures and Tables

**Figure 1 nanomaterials-13-01240-f001:**
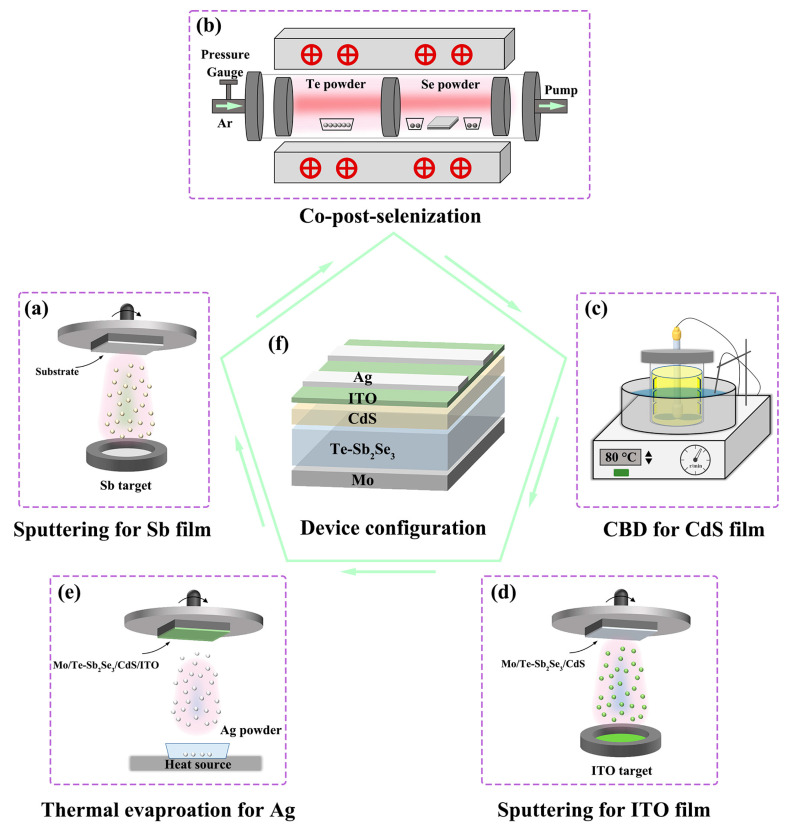
Schematic diagram of the preparation process of the Sb_2_Se_3_ solar cell. (**a**) sputtering for Sb thin film. (**b**) Co-post-selenization heat treatment of Te-doped Sb_2_Se_3_. (**c**) Chemical bath deposition (CBD) method deposited CdS buffer layer. (**d**) A magnetron sputter-deposited ITO window layer. (**e**) Thermal evaporation for Ag electrode. (**f**) The schematic layout of the completed Te-doped Sb_2_Se_3_ device.

**Figure 2 nanomaterials-13-01240-f002:**
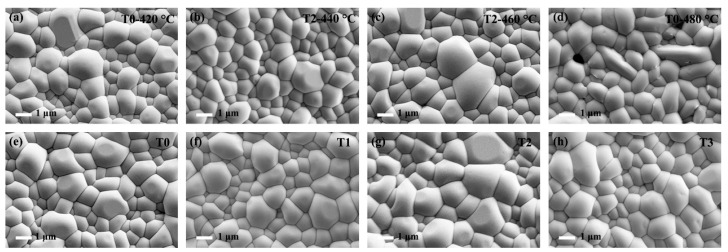
Surface morphology image of the pure Sb_2_Se_3_ and Te-Sb_2_Se_3_ thin films under different selenization temperatures, denoted as (**a**) T0-420 °C, (**b**) T2-440 °C, (**c**) T2-460 °C, and (**d**) T2-480 °C, respectively. Top-view SEM picture of the pristine Sb_2_Se_3_ and Te-Sb_2_Se_3_ thin films under the different weights of the Te source, labeled as (**e**) T0, (**f**) T1, (**g**) T2, and (**h**) T3, respectively.

**Figure 3 nanomaterials-13-01240-f003:**
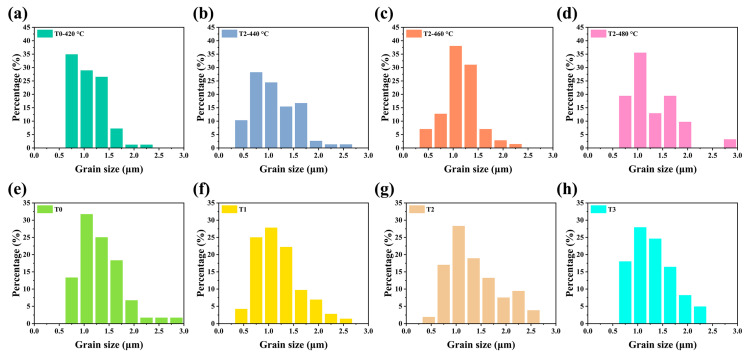
The particle size distribution versus the fine frequency histograms of the pristine Sb_2_Se_3_ and Te-Sb_2_Se_3_ thin films under different selenization temperatures are denoted as (**a**) T0-420 °C, (**b**) T2-440 °C, (**c**) T2-460 °C, and (**d**) T2-480 °C, respectively. (**e**–**h**) The particle size distribution versus the fine frequency histograms of the pristine Sb_2_Se_3_ and Te-Sb_2_Se_3_ thin films under the different weights of the Te source, labeled as (**e**) T0, (**f**) T1, (**g**) T2, and (**h**) T3, respectively.

**Figure 4 nanomaterials-13-01240-f004:**
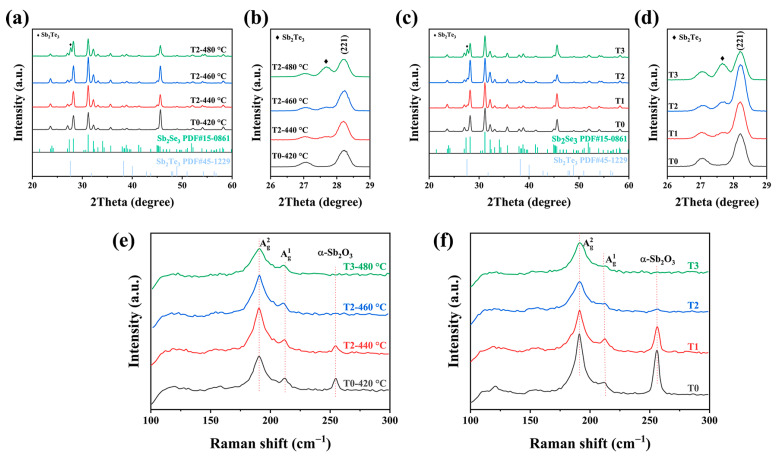
(**a**) Virgin Sb_2_Se_3_ and Te-Sb_2_Se_3_ thin film XRD patterns at various selenization temperatures, and (**b**) enlarged region of the XRD patterns. (**c**) XRD patterns of the pristine Sb_2_Se_3_ and the Te-Sb_2_Se_3_ thin films at different weights of Te source, and (**d**) enlarged region of the XRD pattern. (**e**) Raman spectroscopy of the virgin Sb_2_Se_3_ and the Te-Sb_2_Se_3_ thin films under different selenization temperatures. (**f**) Raman spectroscopy of the virgin Sb_2_Se_3_ and the Te-Sb_2_Se_3_ thin films under different weights of the Te source.

**Figure 5 nanomaterials-13-01240-f005:**
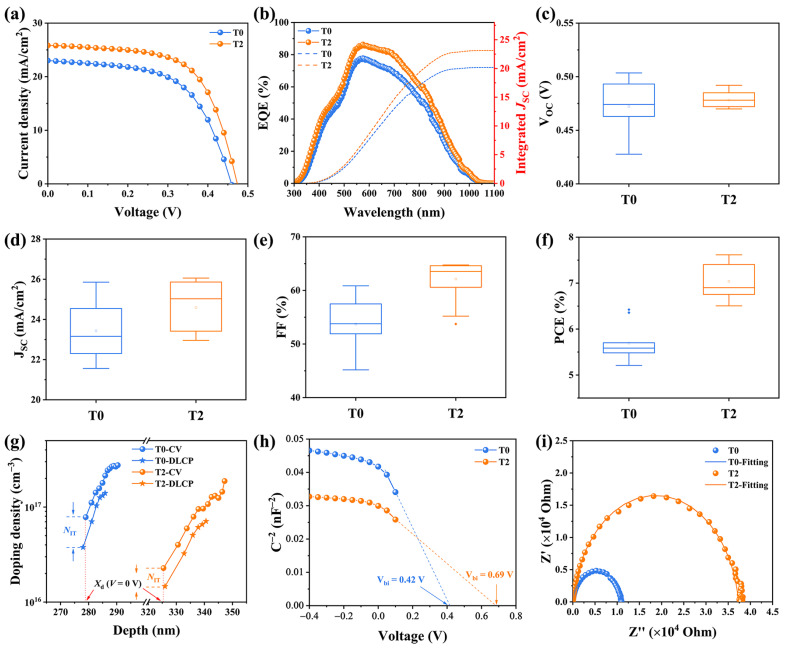
(**a**) The *J*-V curves, and (**b**) the EQE spectroscopy of the T0 and T2 devices. Statistical distribution of the photovoltaic parameter of the T0 and T2 devices, involving (**c**) *V*_OC_, (**d**) *J*_SC_, (**e**) FF, and (**f**) PCE. (**g**) CV and DLCP characterization of the T0 and T2 devices, and (**h**) Plots of C^−2^ against V. (**i**) Nyquist plot of the T0 and T2 devices.

**Figure 6 nanomaterials-13-01240-f006:**
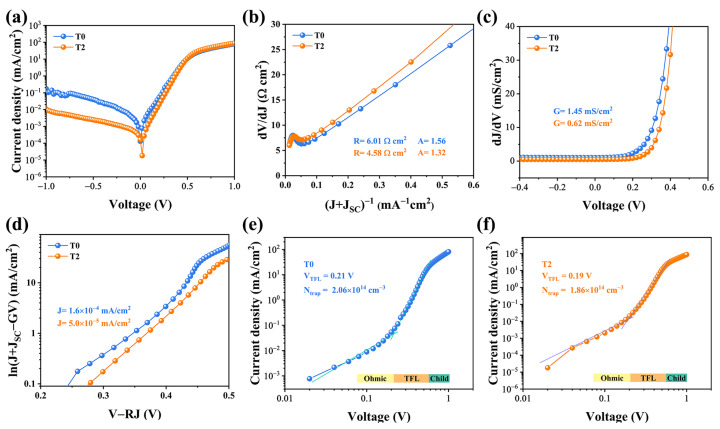
Electrical characteristics of the T0 and T2 devices include the following: (**a**) dark *J*-V curves; (**b**) series resistance R and ideality factor A; (**c**) shunt conductance G; and (**d**) reverse saturation current density *J*_0_. (**e**,**f**) The corresponding logarithmic *J*-V curves of the T0 and T2 devices.

**Figure 7 nanomaterials-13-01240-f007:**
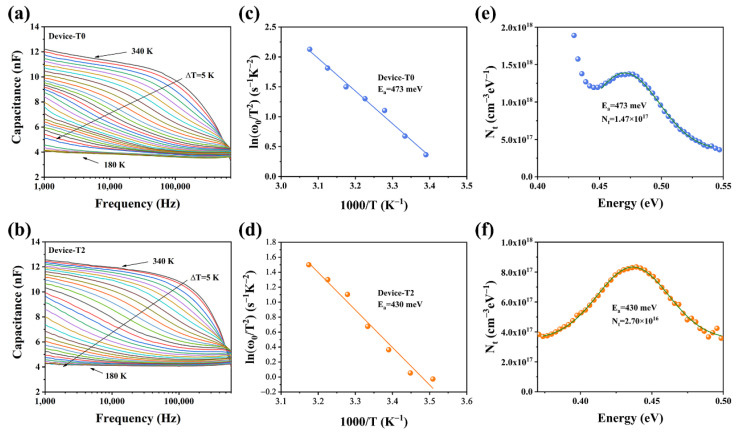
(**a**,**b**) Capacitance−frequency−temperature (C−f−T) spectra (different colored lines represent capacitance values at different temperatures) of the T0 and T2 devices, respectively; (**c**,**d**) Arrhenius plots of the T0 and T2 devices; and (**e**,**f**) defect densities of T0 and T2 devices, respectively.

**Table 1 nanomaterials-13-01240-t001:** Elemental composition of the Sb_2_Se_3_ thin films deposited at the different doping condition (temperature).

Sample	Sb (%)	Se (%)	Te (%)
T0-420 °C	60.54	39.46	0
T2-440 °C	59.99	38.83	1.18
T2-460 °C	39.04	59.06	1.90
T2-480 °C	36.54	61.37	2.09

**Table 2 nanomaterials-13-01240-t002:** Elemental composition of the Sb_2_Se_3_ thin films deposited at the different doping condition (Te weight).

Sample	Sb (%)	Se (%)	Te (%)
T0	38.15	61.85	0
T1	37.67	61.70	0.63
T2	39.17	59.83	0.99
T3	37.10	61.59	1.31

**Table 3 nanomaterials-13-01240-t003:** Device characteristics of the Sb_2_Se_3_ solar cells with or without Te doping.

Device	*V*_OC_ (mV)	*J*_SC_ (mA/cm^2^)	FF (%)	PCE (%)
T0	460	23.15	60.86	6.42
T2	474	25.88	64.09	7.61

## Data Availability

The data presented in this study are available on request from the corresponding authors.

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
