# Peer review of "Tellurium Doping Inducing Defect Passivation for Highly Effective Antimony Selenide Thin Film Solar Cell"

_nanomaterials, 2023, doi:10.3390/nano13071240_

Round 1

Reviewer 1 Report

Referee’s report on the manuscript entitled

“Tellurium Doping Inducing Defects Passivation for Highly Efficient

Antimony Selenide Thin Film Solar Cell”

By Guojie Chen  et al.

The authors report on the growth of antimony selenide (Sb2Se3) thin films by selenization treatment and on tellurium doping during the growth. The films were characterized by various experimental methods, which showed that as a result of the tellurium doping, inclusions of Sb2Te3 are formed in Sb2Se3 instead of the expected solid solution Sb2(Se,Te)3. The authors claim that the tellurium doping leads to mitigation of surface oxidation and influences the orientation of the grains, leading to better charge carrier transport.

The second part of the manuscript describes the fabrication and characterization of Mo/Sb2Se3:Te/CdS/Ito/Ag solar cells. Based on the results, the authors claim that tellurium doping improves the performance of such devices. The manuscript is well-written and contains many details on the fabrication and characterization of the films and solar cells studied. It deserves to be published, however, some questions need to be answered and some issues addressed before publication.

These questions/issues are

1.         Change in the title from "Defects" to "Defect".

2.         Consider changing "Highly Efficient" to "Highly Effective".

3.         Make all Figures readable

4.         How was the active area of the solar cells (0.12 cm2) defined (by lithography or what)?

5.         Change the order in Fig. 1 - the CdS film was deposited before the ITO film was sputtered, so it should be in Fig. 1c and the latter in Fig. 1d.

6.         Write how thick the Sb2Se3:Te and CdS layers are in the solar cells.

7.         What is the type of conductivity and carrier concentration in the Sb2Se3 and Sb2Se3:Te layers?

8.         From what point is the depth measured in Fig. 5g?

9.         How do you interpret the boxes in Fig. 5c-f?

10.       How many devices  were used for the statistics shown in Figures 5c-f?

Author Response

Response to Reviewer #1:

We are thankful to the reviewer for raising concerns over several important points to improve the quality of our manuscript. We have addressed all the points raised with point-by-point responses.

The authors report on the growth of antimony selenide (Sb2Se3) thin films by selenization treatment and on tellurium doping during the growth. The films were characterized by various experimental methods, which showed that as a result of the tellurium doping, inclusions of Sb2Te3 are formed in Sb2Se3 instead of the expected solid solution Sb2(Se,Te)3. The authors claim that the tellurium doping leads to mitigation of surface oxidation and influences the orientation of the grains, leading to better charge carrier transport.

The second part of the manuscript describes the fabrication and characterization of Mo/Sb2Se3:Te/CdS/Ito/Ag solar cells. Based on the results, the authors claim that tellurium doping improves the performance of such devices. The manuscript is well-written and contains many details on the fabrication and characterization of the films and solar cells studied. It deserves to be published; however, some questions need to be answered and some issues addressed before publication.

These questions/issues are.

  1. Change in the title from "Defects" to "Defect".

Author reply:

Thanks for your suggestions. We have revised the manuscript title as follows: Tellurium Doping Inducing Defect Passivation for Highly Effective Antimony Selenide Thin Film Solar Cell.

  1. Consider changing "Highly Efficient" to "Highly Effective"

Author reply:

Thanks for your suggestions. We have revised the manuscript title as follows: Tellurium Doping Inducing Defect Passivation for Highly Effective Antimony Selenide Thin Film Solar Cell.

  1. Make all Figures readable.

Author reply:

Thanks for your suggestions. We have revised it.

  1. How was the active area of the solar cells (0.12 cm2) defined (by lithography or what)?

Author reply:

Thanks for your suggestions. We use a custom mask to evaporate the electrode and then subtract the area of the electrode from the total area to get the effective area of the device.

  1. Change the order in Fig. 1 the CdS film was deposited before the ITO film was sputtered, so it should be in Fig. 1c and the latter in Fig. 1d.

Author reply:

Thanks for your suggestions. We have revised it in the manuscript.

  1. Write how thick the Sb2Se3:Te and CdS layers are in the solar cells.

Author reply:

Thanks for your suggestions. The thickness of the Sb2Se3:Te and the CdS layers are estimated about to 1.2~1.5 μm and 70~100 nm.

  1. What is the type of conductivity and carrier concentration in the Sb2Se3 and Sb2Se3:Te layers?

Author reply:

Thanks for your suggestions. In previous work, (Adv. Mater. 2022, 2109078) has reported the UPS data for the Sb2Se3 solar cell, which is obtained by sputtering Sb thin film and then selenization. In this work, we introduced the same process to prepare the Sb2Se3 and Sb2Se3:Te thin film. In addition, Sun. et al. (Nano Energy 2023,107, 108176) demonstrated the P type Sb2Te3 thermoelectric. So that we believe that the conductivity of the Sb2Se3 and Sb2Se3:Te layers are both P type. We believe that the introduction of small amounts of tellurium will not modify the conductivity of the thin film, but it can optimize the quality of the film to some extent. Besides, as can be seen in Fig. 5h that the Vbi has increased after Te doping, which suggested that the increasing carrier concentration compared that of the pristine Sb2Se3.

  1. From what point is the depth measured in Fig. 5g?

Author reply:

Thanks for your suggestions. The depth can be calculated form the CV profiling at zero bias voltage based on the equation in the manuscript.

  1. How do you interpret the boxes in Fig. 5c-f?

Author reply:

Thanks for your suggestions. The boxplot can more intuitively show the repeatability of the device and the variation of the performance parameters. As depicted in Fig. 5c-f, notably, the enhancement of the performance after Te-doping could be mainly due to the synergetic development of the photovoltaic parameters, confirming that Te-doping may enhance the efficiency of the carrier transport, leading to better thin films.

  1. How many devices were used for the statistics shown in Figures 5c-f?

Author reply:

Thanks for your suggestions. There are fifteen devices used for each statistic shown in Fig. 5c-f.

Reviewer 2 Report

The authors present an interesting paper of tellurium doped antimony selenide synthesis and analysis of its properties. The paper is well written and includes data from most of the commonly used (and required) analysis method. Accordingly I recommend to accept the manuscript after correction of few minor issues:

- Fig. 1 has a misplacement of images 1c and 1d.

- Presenting and discussing the composition of the samples I would suggest to use a term of "elemental composition" instead of "chemical composition". The later term would be more suitable for the chemical phase analysis which is not presented in current study.

- There are no information what method authors used to measure elemental composition and accordingly it is not clear what can be the composition measurement errors (for data presented in tables 1-2)

- Authors provide some comments and discussions on surface oxidation and oxygen containing phases, however, in tables 1 and 2 there are not data for oxygen. It is suggested to review the presentation of this data.

- sentences at lines 17-18, top of page 9, can be reviewed for better English.

- Sample morphology at line 140 is described as "flat surface", however, it is not the best description of visible SEM images.

- Though, presented abbreviations are well known for the specialists in the field, still it would be suggested to include and full wording at their first usage.

Author Response

Response to Reviewer #2:

Thank you for your positive response and recommendation to accept our work in Nanomaterials. Herein, we tried our best to improve the manuscript according to your suggestions.

The authors present an interesting paper of tellurium doped antimony selenide synthesis and analysis of its properties. The paper is well written and includes data from most of the commonly used (and required) analysis method. Accordingly, I recommend accepting the manuscript after correction of few minor issues:

Fig. 1 has a misplacement of images 1c and 1d.

Author reply:

Thanks for your suggestions. We have revised it and the updated image has been put in the manuscript.

Presenting and discussing the composition of the samples I would suggest to use a term of "elemental composition" instead of "chemical composition". The later term would be more suitable for the chemical phase analysis which is not presented in current study.

Author reply:

Thanks for your suggestions. We have revised it in the manuscript.

There is no information what method authors used to measure elemental composition and accordingly it is not clear what can be the composition measurement errors (for data presented in tables 1-2)

Author reply:

Thanks for your suggestions. The elemental composition was measured via the energy-dispersive X-ray spectroscopy (EDS).

Authors provide some comments and discussions on surface oxidation and oxygen containing phases, however, in tables 1 and 2 there are not data for oxygen. It is suggested to review the presentation of this data.

Author reply:

Thanks for your suggestions. We preliminarily believe that this oxidation phenomenon occurs on the surface. For Raman measurement, the detection depth is not large, so surface oxidation phenomenon can be detected. However, compared with EDS measurement, the detection depth is significantly larger than Raman measurement, so the oxygen content in the bulk is relatively negligible.

Sentences at lines 17-18, top of page 9, can be reviewed for better English.

Author reply:

Thanks for your suggestions. We have reviewed it in the manuscript.

Sample morphology at line 140 is described as "flat surface", however, it is not the best description of visible SEM images.

Author reply:

Thanks for your suggestions. We have revised it. "All the as-obtained films displayed a relatively dense surface morphology."

Though, presented abbreviations are well known for the specialists in the field, still it would be suggested to include and full wording at their first usage.

Author reply:

Thanks for your suggestions. We have revised it.

Reviewer 3 Report

The work entitled “Tellurium Doping Inducing Defects Passivation for Highly Efficient Antimony Selenide Thin Film Solar Cell” is original and interesting. However, some recommendations should be taken into account for publication:

Page 4 Line 119-120 - The authors wrote: “The Te-Sb2Se3 thin films selenized with the different weights of Te powder were denoted as T0, T1, T2, and T3, respectively”. But what is the weight of Te powders denoted as T0, T1, T2 and T3?

Consider whether the scale in Figure 3 c should not be as on the “Percentage” axis in Figures 3a,b and d.

Author Response

Response to Reviewer #3:

Thank you for your positive response and recommendation to accept our work in Nanomaterials. Herein, we tried our best to improve the manuscript according to your suggestions.

The work entitled “Tellurium Doping Inducing Defects Passivation for Highly Efficient Antimony Selenide Thin Film Solar Cell” is original and interesting. However, some recommendations should be taken into account for publication:

Page 4 Line 119-120 - The authors wrote: “The Te-Sb2Se3 thin films selenized with the different weights of Te powder were denoted as T0, T1, T2, and T3, respectively”. But what is the weight of Te powders denoted as T0, T1, T2 and T3?

Author reply:

Thanks for your suggestions. We have made corresponding explanations in the manuscript. “T0 is correspond to the weight of 0 g Te powder, T1 is correspond to the weight of 0.1 g Te powder, T2 is correspond to the weight of 0.15 g Te powder and T3 is correspond to the weight of 0.2 g Te powder.”

Consider whether the scale in Figure 3 c should not be as on the “Percentage” axis in Figures 3a,b and d.

Author reply:

Thanks for your suggestions. We have put the updated image in the manuscript.
